# Estimation of Cavities beneath Plate Structures Using a Microphone: Laboratory Model Tests

**DOI:** 10.3390/s21092941

**Published:** 2021-04-22

**Authors:** Seonghun Kang, Jung-Doung Yu, Won-Taek Hong, Jong-Sub Lee

**Affiliations:** 1School of Civil, Environmental and Architectural Engineering, Korea University, Seoul 02841, Korea; gshnice@korea.ac.kr (S.K.); noorung2@korea.ac.kr (J.-D.Y.); 2Department of Civil & Environmental Engineering, Gachon University, Seongnam 13120, Korea; wthong@gachon.ac.kr

**Keywords:** cavity, flexibility, MARSE energy, microphone, sound waves, wavelet transform

## Abstract

The objective of this study is to detect a cavity and estimate its size using sound waves in a laboratory model chamber filled with dry sand. One side of the chamber is covered with an acrylic plate, and a cavity is placed between the plate and sand. Sound waves are generated by impacting the plate with an instrumented hammer, and are measured using a microphone. The measured sound waves are analyzed with four comprehensive analyses including the measured area under the rectified signal envelope (MARSE) energy, flexibility, peak magnitude of wavelet transform, and frequency corresponding to the peak magnitude. The test results show that the accuracy of cavity detection using the MARSE energy is higher for thicker plates, whereas that using flexibility is higher for thinner plates. The accuracies of cavity detection using the peak magnitude of wavelet transform, and frequency corresponding to the peak magnitude are consistently high regardless of the plate thickness. Moreover, the cavity size may be under- or overestimated depending on the plate thickness and the selected analysis method. The average of the cavity sizes estimated by these methods, however, is slightly larger than the actual cavity size regardless of the plate thickness. This study demonstrates that microphones may be effectively used for the identification of a cavity and the estimation of its size.

## 1. Introduction

In urban areas, cavities may occur beneath plate structures such as asphalt or concrete pavements, retaining walls, or underground utilities due to damaged buried pipes or groundwater leakage during underground construction [1]. Damage to buried pipes may lead to leakage of water, which washes out soil particles and produces cavities. Furthermore, soil particles near the damaged pipe flow into the pipe, thereby loosening the surrounding subgrade and inducing a cavity. Cavities beneath the plate structure are difficult to detect with the naked eye [2], and they may lead to rapid failure, which may in turn result in significant damages [3].

Ground penetrating radar (GPR) and surface wave surveys are nondestructive methods commonly used to investigate wide underground areas [1,4,5,6]. A GPR survey investigates an underground area by analyzing electromagnetic waves reflected from boundaries marking different electrical impedances of the media [7,8]. Hong et al. [1] compared the polarity of a GPR signal and the relative density of a subgrade for the estimation of the size and depth of a loose layer or cavity in urban areas. Hong and Lee [5] evaluated the depth, length, and roof shape of cavities by calculating the coordinates of the reflection points of GPR signals. GPR surveys are suitable for the investigation of cavities located at relatively shallow depths. However, GPR signals may be overlapped by strong multiple reflections caused by buried metal objects, or electrical noise caused by high-voltage electric wires [9,10,11]. Yang et al. [10] demonstrated that the strong reflection from rebar was the main obstacle in cavity detection, and suggested the regression method to improve the analysis accuracy in GPR. Thus, GPR surveys are not suitable to be applied in urban areas where several steel pipes and high-voltage electric wires are buried.

In a surface wave survey, which may minimize the effects of buried metal objects, the presence of cavities is evaluated by analyzing the surface wave profile of the underground area. The area where the dispersion curve varied from that of a normal area was distinguished as a cavity section [12,13]. Previous studies demonstrated that surface waves are a useful indicator to identify the presence of cavity. For a surface wave survey, ground-coupled receivers have been widely adopted. However, ground-coupled receivers should be attached in advance, and significant time is required for surface wave detection. Hu et al. [14] showed that inaccurate data caused by poor contact of receivers with the ground could deteriorate the analysis accuracy. Thus, differences in coupling between the receiver and surface should be minimized across the entire measurement location.

By contrast, an air-coupled receiver enables the rapid measurement of the surface wave without the attachment process. Information regarding the integrity of the structure is generally analyzed using signals in the low frequency range, which is below 100 kHz [15]. Among several types of in-air receivers, ultrasonic transducers efficiently generate and transmit ultrasonic energy in the frequency range of 50 kHz to 1 MHz [16]. On the other hand, microphones can sensitively measure signals in the low frequency range (up to 25 kHz) with a flat frequency response and minor attenuation [17]. In addition, microphones are used over a relatively wide bandwidth compared to other in-air receivers such as piezoelectric condensers [15]. Note that microphones have been used to detect shallow cavities located inside or beneath plate structures [18,19]. Thus, a microphone was adopted in this study to rapidly investigate the cavity beneath the plate structure.

According to previous studies, cavity detection has commonly been performed with a single analysis of sound waves such as the MARSE energy [20] or the main frequency [19]. In this study, laboratory tests were performed on a small-scaled model with a cavity. The generation and measurement of sound waves were accomplished using an instrumented hammer and a microphone, respectively. The measured sound waves were analyzed with four comprehensive analyses including the measured area under the rectified signal envelope (MARSE) energy in the time domain, flexibility in the frequency domain, and wavelet transform in the time-frequency domain to identify the cavity and estimate its size. This study includes the measurement system of sound waves, experimental setup, analysis methods, and analysis results of signals in the aforementioned domains, and a summary and conclusions.

## 2. Experimental Study

### 2.1. Measurement System

Sound waves were generated by impacting a modeled acrylic plate using an instrumented hammer (086D05, PCB Piezotronics, Depew, NY, USA), as shown in Figure 1. A load cell was embedded in the instrumented hammer. Thus, the load applied to the acrylic plate at the time of impact could be calculated based on the maximum amplitude of the measured load signal. A plastic tip was attached to the hammer to serve as a cushion at the moment of impact. The diameters of the hammer and plastic tip were 25 mm and 6.3 mm, respectively, and the weight of the hammer was 10.8 N.

Sound waves generated upon hammer impact were measured using a microphone (KB40, K-bell, Seoul, Korea) with a diameter of 12 mm, as shown in Figure 1. The sensitivity of the microphone used in this study is 14 mV/Pa and the microphone has a flat frequency response up to 10 kHz. The signal-to-noise ratio of the microphone is 10.8 dB. Information about the mechanical properties of the plate and the underlying medium is mainly included in the leaky surface wave, rather than in the direct in-air sound wave. The accuracy of the sound wave analysis can be improved by increasing the relative amplitude of the leaky surface wave to the direct in-air sound wave using a rigid or flexible cup-shaped waveguide [21]. In this study, a cup-shaped rubber waveguide was attached to the lower part of the microphone. The dimensions of the waveguide are 14 mm in diameter, 12 mm in height, and 2 mm in thickness. Furthermore, the waveguide served as a positioner for placing the microphone at a specific height, and the microphone height could be adjusted by changing the waveguide height. After the hammer impact, the load signal generated by the instrumented hammer, and the sound wave measured by the microphone were filtered and amplified, represented, and recorded by a filter-amplifier (3944, Krohn-Hite Corporation, Brockton, MA, USA), an oscilloscope DSOX3014A, Keysight Technologies, Santa Rosa, CA, USA), and a laptop computer, respectively, as shown in Figure 1. The signals were acquired at a sampling rate of 625 kHz, and the number of sampling points was 4000.

### 2.2. Experimental Setup

A chamber containing soil and a model cavity was prepared, as shown in Figure 2, to simulate a cavity beneath the asphalt pavement. The asphalt pavement was modeled using an acrylic plate because the elastic modulus of the acrylic plate is similar to that of asphalt pavement (approximately 3 GPa) [22]. Figure 2a–c shows the front view of the chamber, the side view in section B–B’, and the bottom view in section A–A’, respectively. The dimensions of the chamber are 800 mm in length, 800 mm in height, and 300 mm in width. A square acrylic plate with 800 mm sides was installed in front of the chamber as shown in Figure 2b. Two acrylic plates with thicknesses of 8 mm and 15 mm (T = 8 and 15 mm) were used to evaluate the effect of the plate thickness.

The chamber was filled using dry sand with a median particle diameter (D_50_) of 0.63 mm, specific gravity of 2.65, and friction angle of 36°. The dry sand was compacted in the chamber at a relative density of 64%. The compaction was carried out by dropping a hammer with a weight of 24.5 N and a diameter of 50 mm. The hammer was dropped 50 times for each layer from a height of 500 mm. The thickness of each layer was 100 mm, and the chamber was prepared in eight layers. The cavity was simulated by placing a plastic box because the distance between the measurement location and the edge of the cavity can be clearly defined for a scan line (one-dimensional) test [19]. The dimensions of the cavity are 120 mm in length, 100 mm in width, and 180 mm in height between the acrylic plate and the soil, as shown in Figure 2a,b. The cavity was positioned at the center of the acrylic plate as shown in Figure 2a.

## 3. Experimental Results

### 3.1. Height of Microphone and Spacing between Hammer and Microphone

During the measurement of sound waves using an air-coupled microphone, the signal is affected by the microphone height and the center-to-center spacing between the hammer and microphone [15,23]. As the air-coupled microphone height from the ground increases, the main frequency of the sound wave decreases relative to that measured using the ground-coupled sensor [23]. In addition, as the center-to-center spacing between the hammer and microphone becomes narrower, the inspection area by the microphone becomes narrower, and the spatial resolution of the measurement result increases [15]. The sound waves were measured while adjusting the microphone height (h) and the center-to-center spacing between the hammer and microphone (s) to determine the optimal values for each variable. An experiment for determination of h and s was performed on the B–B’ line, at the cavity section with a 15 mm thick acrylic plate, as shown in Figure 3. The measured sound wave in units of mV was normalized by the load in units of N measured by the instrumented hammer at each impact to remove the effect of the impact energy.

To determine the effect of the microphone height on the measured sound wave, the sound waves were measured while maintaining the spacing (s) constant and adjusting only the height (h). Considering the diameter of the hammer and the waveguide (25 mm and 14 mm, respectively), the minimum value of s was 19.5 mm (i.e., when the hammer and the waveguide were in contact). Thus, the initial fixed value of s was set as 20 mm for convenience as shown in Figure 3, and the sound waves were measured by adjusting h from 5 to 25 mm in 5 mm increments (Figure 3b). For the height (h) determination tests, the first wave crest and the first wave trough (C_1_ and T_1_, respectively in Figure 4a) appeared at approximately 0.06 ms and 0.5 ms, respectively. The normalized peak-to-peak amplitudes between C_1_ and T_1_ according to the microphone height are plotted in Figure 4b. Note that at the h = 5 mm, the value of T_1_ was saturated as shown in Figure 4a. At h = 10 mm, the sound wave was clearly measured without saturation, and thus the microphone height was fixed to 10 mm in this study.

The sound waves were measured by adjusting s from 20 to 60 mm in 10 mm increments under a fixed h = 10 mm (Figure 3b). The experimental results (Figure 4c) show that saturation of the sound wave did not occur at all values of s. As the maximum peak-to-peak amplitude was measured at s = 20 mm (Figure 4d), the center-to-center spacing between the hammer and microphone was set to s = 20 mm under h = 10 mm.

When the hammer vertically impacts the thin plate, a Lamb wave is generated. The wavelength of the Lamb wave is significantly longer than the plate thickness, and the lowest order (S0 and A0) modes appear [24]. As the A0 mode is predominant compared to the S0 mode for the vertical impact [25], the A0 Lamb wave mode was used in this study. The group velocity of the Lamb wave (v_L_) on the acrylic plate is calculated as follows:v_L_ = s/(t_C_ − t_a_)(1)
where s is the center-to-center spacing between the hammer and microphone (20 mm); t_C_ is the travel time (≈0.06 ms) of the Lamb wave; and t_a_ is the travel time (≈0.03 ms) of the sound wave transmitted through the air at a velocity of 343 m/s from the plate surface to the microphone height (10 mm). v_L_ was calculated to be approximately 667 m/s from Equation (1). The dispersion curve of the Lamb wave according to the multiplication of frequency and thickness in several polymer materials showed that the group velocity of the A0 mode Lamb wave in the far field was 800–900 m/s [26]. The measured Lamb wave in this study, however, was approximately 667 m/s due to the near field effects [27]. Yoon and Rix [27] showed that the Lamb wave velocity in the near field is 0.6–0.8 times that in the far field. Note that the center-to-center spacing between the hammer and microphone (s) was set as 20 mm. Figure 4a shows that, the second wave crest (C_2_) appears at approximately 1.2 ms. The travel length of C_2_ under v_L_ = 667 m/s was calculated as approximately 800 mm, which is the round-trip path from the hammer impact point to the chamber boundary.

### 3.2. Measured Sound Waves

The sound wave measurements were conducted at a total of ten locations (six and four locations at the cavity and soil sections, respectively) along the A–A’ line as shown in Figure 5. The measurement locations at the cavity section were placed from 0 to 50 mm in 10 mm increments from the cavity center (L = 0–50 mm), and those at the soil section were placed 80, 100, 150, and 200 mm from the cavity center. Windowing was applied to the measured sound wave to analyze only the sound wave without the reflected Lamb wave from the boundary. As an example of the windowing application, the raw sound wave, measured at the measurement location of L = 0 mm for the 15 mm thick acrylic plate is shown as the “raw signal” in Figure 6. The Tukey window function (“windowing signal” in Figure 6) was applied to the raw sound wave [28], and the windowed sound wave (“windowed signal” in Figure 6) was obtained by multiplying the raw sound wave and the window function. A window length of 1 ms was applied for all raw signals, and the taper ratio was set to 0.2 as shown in Figure 6.

The raw sound waves at all measurement locations of the two acrylic plates with different thicknesses (T = 8 and 15 mm) are shown in Figure 7a,b. In addition, the windowed normalized sound waves by the hammer impact, i.e., the ratio of the windowed raw sound waves to the peak amplitude of the load signal due to hammer impact, are plotted in Figure 7c,d. According to Figure 7, the maximum peak-to-peak amplitude of the first wave crest and trough appeared at the measurement location of L = 0 mm, for both acrylic plates, and gradually decreased as the measurement location became farther from the cavity center. Thus, the peak-to-peak amplitudes of the first wave crest and trough at the cavity section are larger than that at the soil section.

## 4. Analyses and Discussion

### 4.1. MARSE Energy

#### 4.1.1. Value of MARSE Energy

The MARSE energy, which is the area between the absolute value of the signal and the time axis in units of s·V/N [29,30], is expressed as follows:(2)ME=∫|g(t)|·dt
where ME is the MARSE energy; and *g*(*t*) is the amplitude of the signal with respect to time. Note that the MARSE energy has been used to analyze the attenuation of signal [31]. The MARSE energies calculated for the two acrylic plates are summarized in Table 1 and plotted according to the measurement locations shown in Figure 8. Figure 8 shows that the MARSE energy at the cavity section is higher than that at the soil section for both acrylic plates. When a cavity exists beneath a plate structure, the impact energy is conserved [32]. Furthermore, the estimated MARSE energy of the 15 mm thick acrylic plate was lower than that of the 8 mm thick acrylic plate for all the measurement locations. As the thickness of the plate structure increases, the maximum deflection and dynamic response of the structure generated by the hammer impact decrease [32,33]. Therefore, the maximum amplitude of the sound wave in the time domain decreases [34], and the MARSE energy decreased accordingly with an increase in the plate thickness.

#### 4.1.2. Normalized MARSE Energy

The MARSE energy at all measurement locations was normalized by that at L = 200 mm (soil section). The normalized MARSE energies according to the measurement locations are summarized in Table 1 and plotted in Figure 9a. Figure 9a shows that for the 8 mm acrylic plate, the maximum normalized MARSE energy (approximately 1.8) occurring at the measurement location of L = 0 mm gradually decreased along L = 50–100 mm and remained almost constant after L = 100 mm. For the 15 mm acrylic plate, the maximum normalized MARSE energy (approximately 5.4 at L = 0 mm) gradually decreased along L = 0–30 mm and significantly decreased at L = 30–50 mm. Subsequently, the normalized MARSE energy gradually decreased further along L = 50–100 mm and remained almost constant after L = 100 mm. Figure 9a shows that the normalized MARSE energy at the cavity section increased with increasing plate thickness. When the plate structure was placed on the half-space medium, the maximum deflection and dynamic response of the plate structure decreased with increasing stiffness ratio between the plate structure and the underlying medium [35]. The stiffness ratio (K) is expressed as follows [33,36]:K = (T^3^·E_p_·(1 − ν_medium_))/(12·G_medium_·D^3^·(1 − ν_plate_^2^))(3)
where T is the plate thickness; E_p_ is the elastic modulus of the plate, and ν_medium_ and ν_plate_ are Poisson’s ratios of the medium and the plate, respectively. D is the longest dimension of the plate and G_medium_ is the shear modulus of the underlying medium. At the cavity section, the MARSE energy decreased with increasing plate thickness as shown in Figure 8. However, at the soil section, as the thickness increased, the stiffness ratio increased, and thus the MARSE energy decreased. Thus, the normalized MARSE energy significantly increased with an increase in the plate thickness at the cavity section.

### 4.2. Flexibility

#### 4.2.1. Mobility Spectrum

When the hammer impacts the plate, the Lamb wave propagates along its surface, and the vibration of the plate is transmitted to the molecules (or particles) in the air, thereby generating sound waves [37]. The microphone measures the change in air pressure caused by the sound wave and produces a signal in units of V. A sound wave in volts can be converted to sound pressure (p) in units of Pa as follows:p = MSS/S(4)
where MSS is the measured sound wave in units of V. Note that MSS corresponds to the windowed raw sound signal as represented in Figure 7a,b. S is the sensitivity (14 mV/Pa) of the microphone used in this study. The particle velocity of the sound waves is calculated based on the sound pressure and acoustic impedance of the medium. The particle velocity of air (u) can be calculated by dividing the sound pressure (p) by the acoustic impedance (Z_0_ = 400 N·s/m^3^) of air as follows:u = p/Z_0_ = MSS/(S·Z_0_)(5)

Note that the particle velocity in this study is the vertical movement of a particle of the acrylic plate or air, and thus the particle velocity of the air calculated from the sound wave, which is transmitted from the plate vibration, is proportional to that of the acrylic plate [15].

A plate structure can be represented by a single degree of freedom (SDOF) system [38,39], and the mobility of such an SDOF system can be analyzed using frequency response curves. Mobility is the ratio of the particle velocity (output value) of a structure to the load (input value) applied to the structure [40], and the mobility signal in the frequency domain is the mobility spectrum. For the calculation of the mobility spectrum, the windowed raw sound wave in units of mV was converted to a particle velocity (u) signal in units of m/s using Equation (5). As an example, the windowed raw sound wave measured at L = 0 mm for a 15 mm thick acrylic plate was converted to a particle velocity signal as shown in Figure 10a. The particle velocity and load signals in the time domain (Figure 10a,b, respectively) were converted into signals in the frequency domain (Figure 10c,d, respectively) through the fast Fourier transform (FFT). Mobility is determined by dividing the particle velocity by the load in the frequency domain, and mobility is expressed in units of m/sN [41]. The mobility spectra for all measurement locations of the two acrylic plates are represented in Figure 11. Figure 11 shows that the initial slope of the mobility spectrum decreased as the measurement location becomes farther from the cavity center for both acrylic plates.

#### 4.2.2. Normalized Mobility

Flexibility, which is the slope of the initial part of the mobility spectrum in units of m/N, means the deflection for the applied load [42]. Note that the inverse of flexibility is dynamic stiffness. The flexibilities at all measurement locations for the two acrylic plates were calculated in the frequency range of 0–10 Hz using Figure 11 and summarized in Table 1. In addition, the normalized flexibilities at all measurement locations (normalized by the flexibility at 200 mm, i.e., soil section) are summarized in Table 1 and plotted in Figure 9b. Figure 9b shows that, for the 8 mm acrylic plate, the maximum normalized flexibility (approximately 12.3) was obtained at L = 0 mm and remained almost constant at 0–20 mm. The normalized flexibility significantly decreased along 20–40 mm and continuously decreased beyond 40 mm. For the 15 mm acrylic plate, the maximum normalized flexibility (approximately 3.2) was obtained at L = 0 mm, gradually decreasing along 0–80 mm, and then remained almost constant after 80 mm. The normalized flexibilities at the cavity section were higher than those at the soil section for both acrylic plates because the acrylic plate becomes more flexible at the cavity section due to the loss of support [40]. Furthermore, the difference in the normalized flexibility between the cavity and soil sections increased with decreasing plate thickness. When the hammer impacts the plate structure with a cavity below its surface, a low frequency dominated sound wave may occur due to the flexural vibration behavior of the upper plate [18,19,43]. Note that the area in which a low frequency dominated sound wave was measured was considered as the location of shallow delamination or cavity in concrete slabs [44,45]. As the main frequency at the cavity section was lower than that at the soil section, the low frequency at the cavity is dominant [46]. As the mobility spectrum is calculated by dividing the particle velocity by the load in the frequency domain, the mobility and flexibility increase in the low frequency range. As the flexural vibration due to the hammer impact was greater for the thinner plate (T = 8 mm), the flexibility and the difference in the normalized flexibility between the cavity and soil sections were higher in the thinner plates [46].

### 4.3. Wavelet Transforms

#### 4.3.1. Wavelet Transform Results

The wavelet transform has been widely used to analyze sound waves for the investigation of defects within infrastructures [47,48,49], because the wavelet transform provides the best time-frequency resolution for non-stationary signals [50,51]. In this study, the Gabor wavelet was used as the mother wavelet, and wavelet transforms were conducted using the windowed normalized sound waves shown in Figure 7c,d. The wavelet transform results at measurement locations of L = 0, 20, 40, 80, and 150 mm are plotted in Figure 12.

#### 4.3.2. Peak Magnitude of Wavelet Transform

The magnitude of the wavelet coefficient (*MT_p_*) and the frequency (*fr*) corresponding to the peak magnitude are presented in Figure 12. The peak magnitude and frequency corresponding to the peak magnitude of the wavelet transform for all measurement locations of the two acrylic plate thicknesses are summarized in Table 1. Table 1 shows that the peak magnitudes for the 8 mm thick acrylic plate were greater than those for the 15 mm thick acrylic plate at all measurement locations, because the energy transfer ratio from the hammer impact to the flexural vibration was higher for the thinner plate [18].

#### 4.3.3. Normalized Peak Magnitude

The peak magnitude at all measurement locations was normalized by that at 200 mm (in the soil section), and the normalized peak magnitudes corresponding to the measurement location are summarized in Table 1 and plotted in Figure 9c. The maximum normalized peak magnitude (approximately 3.0) appeared at L = 0 mm and continuously decreased afterward for both acrylic plates. The maximum amplitude of the sound wave increases as the flexural vibration increases [18]. The relationship among the energy transfer ratio from the hammer impact to the flexural vibration (k), Poisson’s ratio of the plate (ν_plate_), plate thickness (T), and elastic modulus of the plate (E_p_) is expressed as follows [52]:k ∝ [(1 − ν_plate_^2^)/(E_p_·T^3^)]^0.5^(6)

As the thickness (T) and properties including the ν and E of the acrylic plates used in this study were uniform, k is constant across all measurement locations for both acrylic plates. The peak magnitudes at all measurement locations indicate the energies of the flexural vibration, which are transferred from the impact through a constant energy transfer ratio (k). Therefore, the effect of k is eliminated by normalization with the value at L = 200 mm (soil section), and thus the normalized peak magnitudes are similar at all measurement locations for both acrylic plates. As the peak magnitudes for both acrylic plates continuously decrease from the center of the cavity, the cavity may be detectable using the peak magnitude.

#### 4.3.4. Frequency Corresponding to Peak Magnitude

The frequencies corresponding to the peak magnitude of the wavelet transform are summarized in Table 1. Table 1 show that the frequency at L = 0 mm was approximately 2.1 kHz and 2.2 kHz for T = 8 mm and T = 15 mm plates, respectively, and increased within 0–80 mm for both acrylic plates. Beyond 80 mm, the frequency merged at 3.4 kHz and 3.5 kHz for the plates with T = 8 mm and T = 15 mm, respectively. The frequency at the cavity section was lower than that at the soil section for both acrylic plates because the main frequency of the flexural vibration decreases at the cavity section [19,46]. Note that as the depth and size of the cavity, and the support condition of the plate are the same for both plates, the frequencies corresponding to the peak magnitude are similar as summarized in Table 1.

#### 4.3.5. Normalized Frequency Corresponding to Peak Magnitude

The frequencies corresponding to the peak magnitude at all measurement locations normalized by that at L = 200 mm (soil section) are summarized in Table 1 and plotted in Figure 9d. Figure 9d shows that the minimum normalized frequency (approximately 0.63) appeared at L = 0 mm for both acrylic plates. Subsequently, the normalized frequency increased linearly within the range of 0–80 mm, and converged at approximately 1.0 beyond 80 mm for both acrylic plates.

### 4.4. Cavity Detection

#### 4.4.1. Cavity Identification

The maximum ratios between the cavity and soil sections for the normalized MARSE energy, flexibility, peak magnitude of wavelet transform, and frequency corresponding to the peak magnitude are summarized in Table 2. The maximum normalized MARSE energies for the 8 mm and 15 mm acrylic plates were 1.8 and 5.4, respectively. The normalized MARSE energy increased with an increase in the plate thickness. Thus, cavity detection using MARSE energy might be more accurate for thicker plates. The maximum normalized flexibilities for the 8 mm and 15 mm acrylic plates were 12.3 and 3.2, respectively. The normalized flexibility increased with a decrease in the plate thickness, therefore, cavity detection using flexibility may be more accurate for thinner plates. As the maximum ratios for the normalized peak magnitude were approximately 3.0 and those for the normalized frequency were approximately 0.63 for both acrylic plates, and thus the accuracy of cavity detection using the peak magnitude and frequency is independent on plate thickness [19,46,52].

#### 4.4.2. Cavity Size Estimation

The cavity size in this study is defined as the location from the cavity center to the boundary between the cavity and soil. For the estimation of the cavity size using the normalized MARSE energy for the 8 mm acrylic plate as shown Figure 9a, the intersection of two lines, i.e., the extended tangent line to the normalized MARSE energy at L = 200 mm, and the extended line for its linear portion within 50–80 mm, was used. The intersection of the two lines was approximately 100 mm, and thus the cavity section was estimated to be 0–100 mm. Considering that the actual cavity section was 0–60 mm, the cavity size was overestimated. Moreover, for the 15 mm acrylic plate, the intersection of the extended tangent line at L = 200 mm and its extended line for the inflection range (40–50 mm) was adopted. Consequently, the cavity size was estimated to be the actual size of 60 mm. The cavity size estimated by the MARSE energy, which is summarized in Table 2, depends on the plate thickness.

For the flexibility (Figure 9b), the intersection of the two extended lines between the linear portion of the normalized flexibility within the ranges of 20–40 mm and 50–200 mm was used as the edge of the cavity for the 8 mm acrylic plate. The estimated cavity size was approximately 40 mm, which was underestimated. Similarly, for the 15 mm acrylic plate, the estimated cavity section was 0–80 mm. Cavity size estimation using the normalized peak magnitude of the wavelet transform was considered to be difficult specifying the cavity edge. For cavity size estimation using the normalized frequency corresponding to the peak magnitude from Figure 9d, the intersection of two lines from the measurement locations of L = 0 mm and 200 mm estimated the cavity section of 0–80 mm for both plates.

The cavity sizes estimated using the four normalized values are summarized in Table 2. Table 2 shows that the cavity size may be under- or overestimated according to the plate thickness and the selected methods. Hence, comprehensive analyses employing the MARSE energy, flexibility, and frequency are essential. As an example of such a comprehensive analysis, the average of the cavity sizes estimated using the MARSE energy, flexibility, and frequency was 73 mm for two plates, which was slightly larger than the actual cavity boundary. Thus, the effect of plate thickness on the estimated cavity size could be minimized by averaging the cavity sizes estimated using the MARSE energy, flexibility, and frequency.

## 5. Summary and Conclusions

To identify the cavity and estimate the cavity size using sound waves, the small-scaled laboratory model tests were conducted in a chamber, of which dimensions were 800 mm in length, 800 mm in height, and 300 mm in width. The chamber was filled with dry sand. One side of the chamber was covered with an acrylic plate to simulate the plate structure. The cavity was simulated using a plastic box, which was placed between the acrylic plate and soil. Two acrylic plates with different thicknesses (T = 8 and 15 mm) were used to evaluate the effect of plate thickness. Sound waves were generated by impacting the acrylic plate with an instrumented hammer, and were measured using a microphone. The microphone height and center-to-center spacing between the hammer and microphone were fixed at 10 mm and 20 mm, respectively. Sound wave measurements were conducted at ten measurement locations (six and four measurement locations at the cavity and soil sections, respectively). The measured sound waves were analyzed with four comprehensive analyses including the measured area under the rectified signal envelope (MARSE) energy in the time domain, flexibility in the frequency domain, and wavelet transform in the time-frequency domain. By normalizing the MARSE energy, flexibility, peak magnitude of wavelet transform, and frequency corresponding to the peak magnitude at all measurement locations by those at location 200 mm (soil section), four normalized values corresponding to different measurement locations were obtained. Those four normalized values were used for cavity detection and cavity size estimation. The main observations obtained from this study were as follows:The (normalized) MARSE energy and the (normalized) flexibility at the cavity section were higher than those at the soil section due to the occurrence of flexural vibration behavior of the plate.Thus, the (normalized) MARSE energy at the cavity section was higher than that at the soil section for both acrylic plates. In addition, because the acrylic plate became more flexible at the cavity section due to the loss of support, the (normalized) flexibility at the cavity section was higher than that at the soil section.The (normalized) peak magnitude of the wavelet transform at the cavity section was greater than that at the soil section due to the higher attenuation at the soil section. Furthermore, the (normalized) frequency corresponding to the peak magnitude at the cavity section was lower than that at the soil section because the main frequency of the sound waves was lower at the cavity section.The accuracy of cavity detection increased for thicker plates with detection using the MARSE energy, and for thinner plates with detection using the flexibility. In addition, the accuracies of cavity detection using both the peak magnitude and the frequency are independent of the plate thickness.Among the four analysis methods, the cavity size can be estimated using the MARSE energy, flexibility, and frequency corresponding to the peak magnitude of the wavelet transform. When the MARSE energy was used, the cavity size was overestimated for the thinner acrylic plate, whereas it was estimated as the actual size for the thicker plate. When the flexibility was used, the cavity size was underestimated for the thinner acrylic plate, and overestimated for the thicker plate. Furthermore, the cavity size was overestimated regardless of the plate thickness based on the frequency. In other words, the cavity size may be under- or overestimated according to the plate thickness and the selected analysis method. On the other hand, the average of the cavity sizes estimated using the three methods was slightly larger than the actual cavity size regardless of the plate thickness. Therefore, the effect of the plate thickness on the estimated cavity size may be minimized by comprehensive analyses of the sound waves.

## Figures and Tables

**Figure 1 sensors-21-02941-f001:**
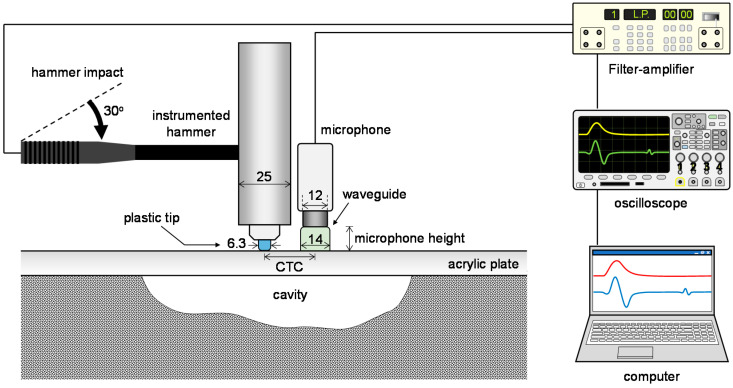
Measurement system. CTC denotes center-to-center spacing between hammer and microphone. Units are given in mm.

**Figure 2 sensors-21-02941-f002:**
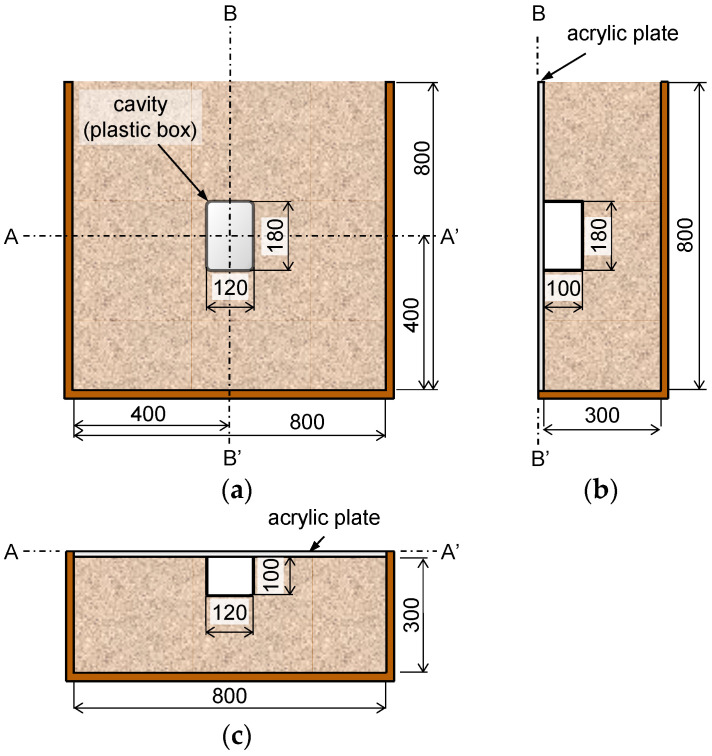
Schematic drawings of chamber: (**a**) front view; (**b**) side view; and (**c**) bottom view. Units are given in mm.

**Figure 3 sensors-21-02941-f003:**
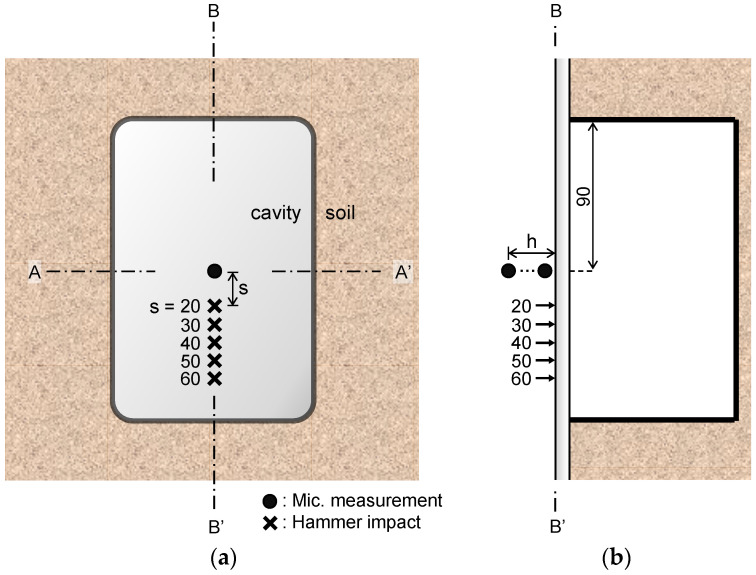
Measurement locations for determination of optimal microphone height and center-to-center spacing between hammer and microphone: (**a**) front view and (**b**) side view. Units are given in mm.

**Figure 4 sensors-21-02941-f004:**
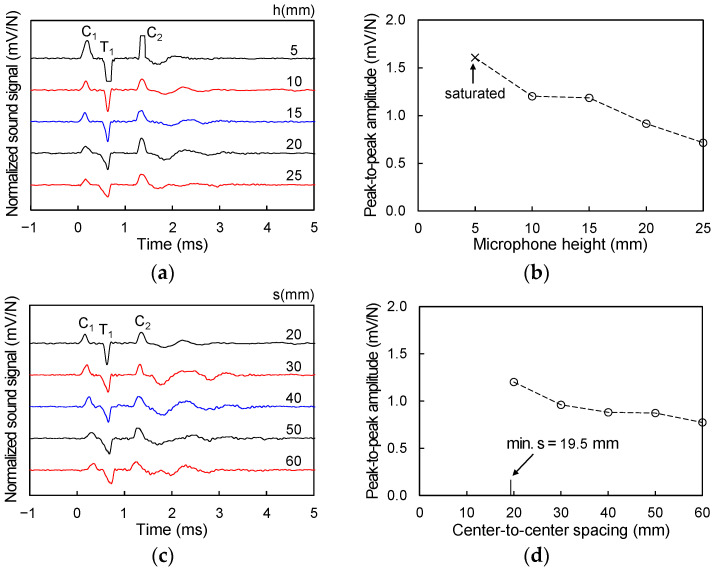
Optimal microphone height and center-to-center spacing between hammer and microphone: (**a**) measured sound waves for different height under fixed spacing (s = 20 mm); (**b**) normalized peak-to-peak amplitude under fixed spacing (s = 20 mm); (**c**) measured sound waves for different spacing under fixed height (h = 10 mm); and (**d**) normalized peak-to-peak amplitude under fixed height (h = 10 mm).

**Figure 5 sensors-21-02941-f005:**
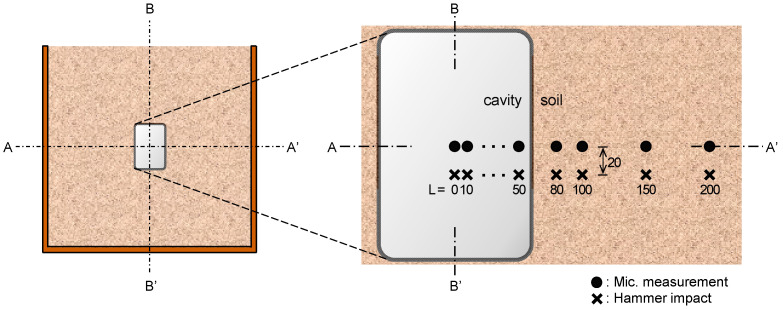
Measurement locations at cavity and soil sections. Units are given in mm.

**Figure 6 sensors-21-02941-f006:**
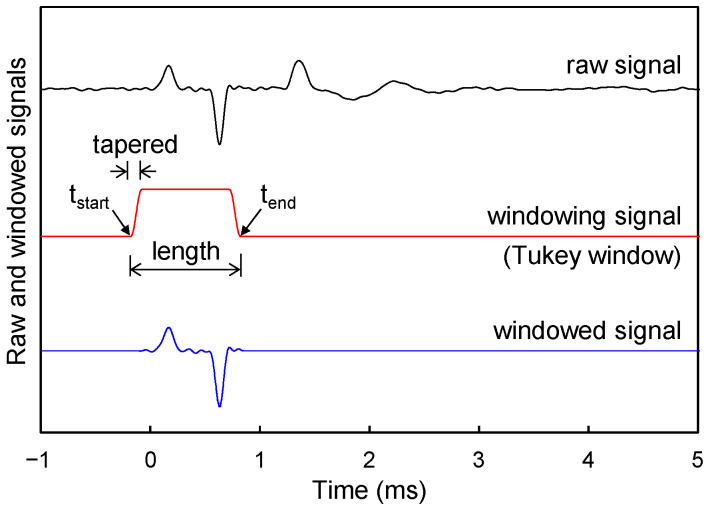
Windowing of sound wave.

**Figure 7 sensors-21-02941-f007:**
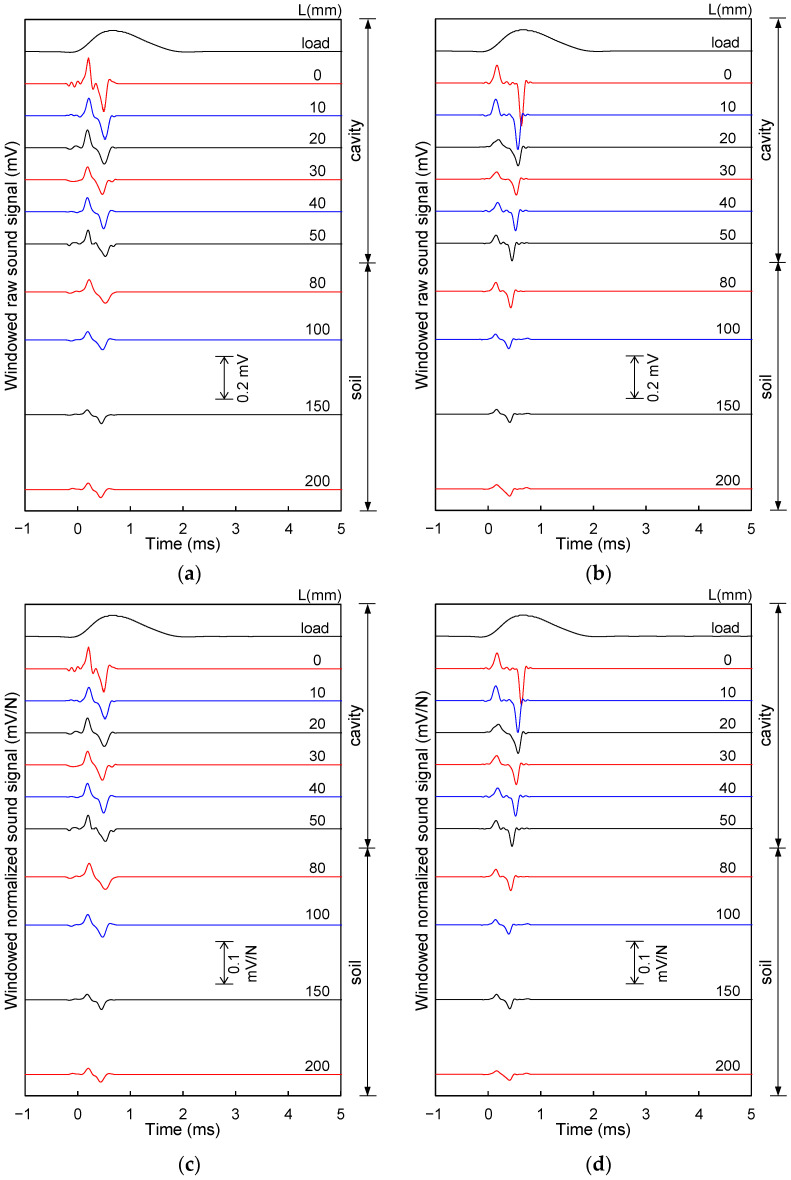
Windowed raw and windowed normalized sound waves according to measurement location: (**a**) windowed raw sound wave for 8 mm thick acrylic plate; (**b**) windowed raw sound wave for 15 mm thick acrylic plate; (**c**) windowed normalized sound wave by hammer impact load for 8 mm thick acrylic plate; and (**d**) windowed normalized sound wave by hammer impact load for 15 mm thick acrylic plate.

**Figure 8 sensors-21-02941-f008:**
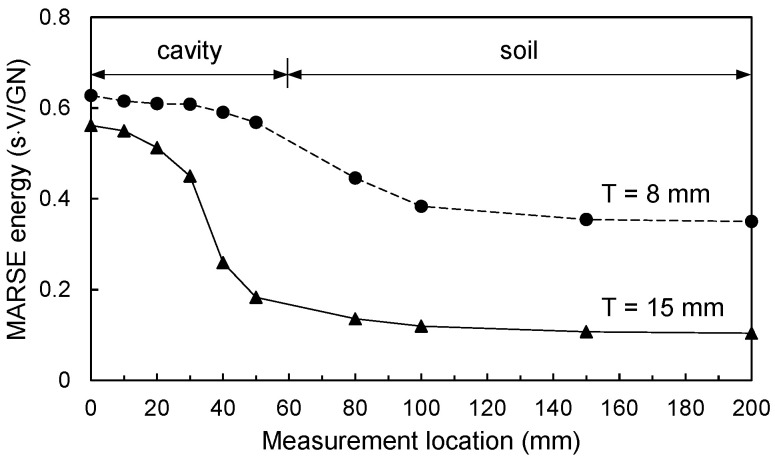
MARSE energies according to measurement location.

**Figure 9 sensors-21-02941-f009:**
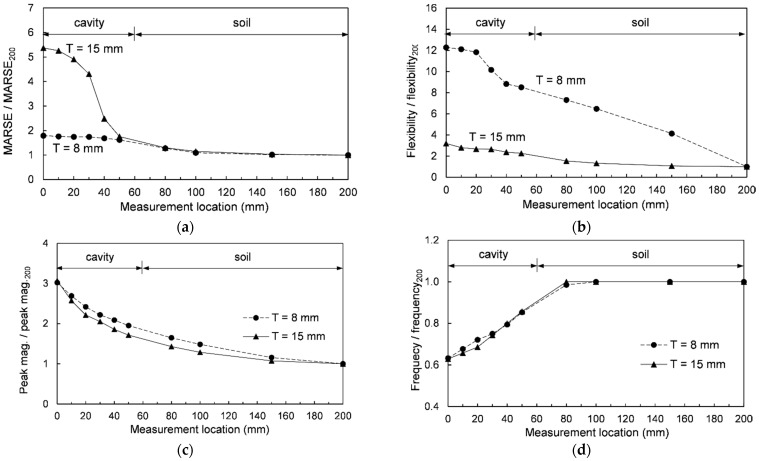
Normalized values according to measurement location: (**a**) MARSE energy; (**b**) flexibility; (**c**) peak magnitude; and (**d**) frequency corresponding to peak magnitude.

**Figure 10 sensors-21-02941-f010:**
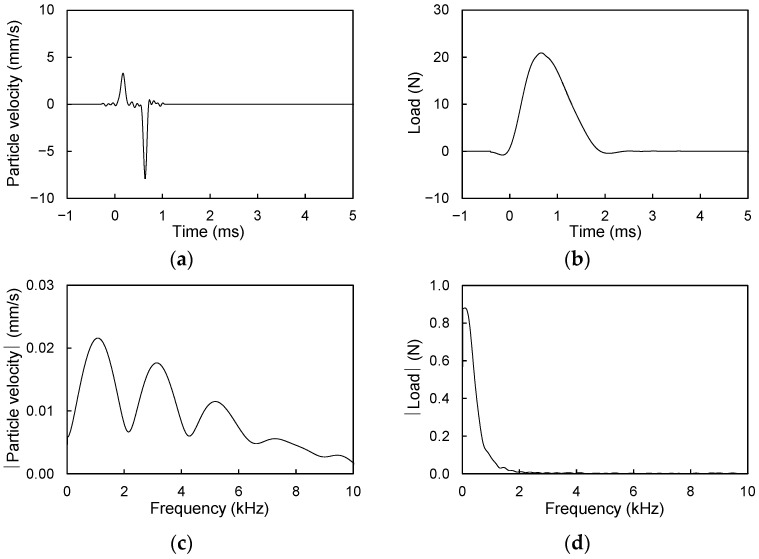
Load and particle velocity signals at measurement location of L = 0 mm for 15 mm thick acrylic plate: (**a**) particle velocity signal in time domain; (**b**) load signal in time domain; (**c**) particle velocity in frequency domain; and (**d**) load signal in frequency domain.

**Figure 11 sensors-21-02941-f011:**
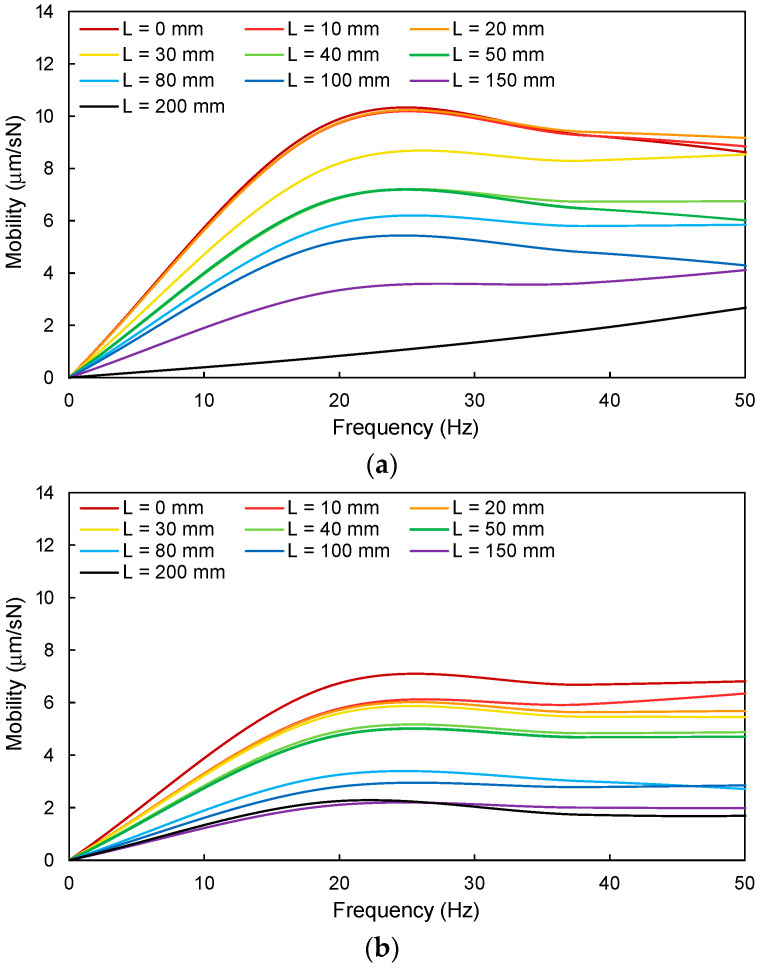
Mobility spectrum according to measurement location: (**a**) 8 mm thick acrylic plate and (**b**) 8 mm thick acrylic plate.

**Figure 12 sensors-21-02941-f012:**
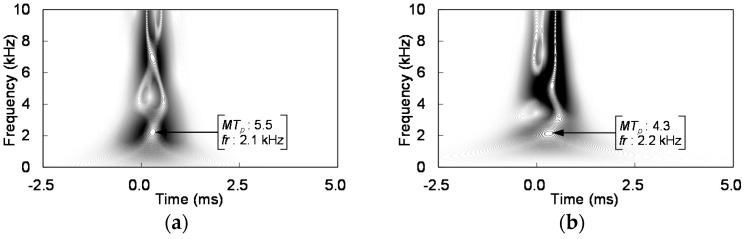
Wavelet transform of sound waves at selected measurement locations: (**a**) L = 0 mm for 8 mm thick acrylic plate; (**b**) L = 0 mm for 15 mm thick acrylic plate; (**c**) L = 20 mm for 8 mm thick acrylic plate; (**d**) L = 20 mm for 15 mm thick acrylic plate; (**e**) L = 40 mm for 8 mm thick acrylic plate; (**f**) L = 40 mm for 15 mm thick acrylic plate; (**g**) L = 80 mm for 8 mm thick acrylic plate; (**h**) L = 80 mm for 15 mm thick acrylic plate; (**i**) L = 150 mm for 8 mm thick acrylic plate; and (**j**) L = 150 mm for 15 mm thick acrylic plate.

**Table 1 sensors-21-02941-t001:** MARSE energy, flexibility, peak magnitude and corresponding frequency of wavelet transform, and their normalized values.

	ME		FL		PM		F		NME		NFL		NPM		NF	
L	8 mm	15 mm	8 mm	15 mm	8 mm	15 mm	8 mm	15 mm	8 mm	15 mm	8 mm	15 mm	8 mm	15 mm	8 mm	15 mm
0	0.63	0.56	508.0	345.7	5.5	4.3	2.1	2.2	1.8	5.4	12.3	3.2	3.1	2.9	0.62	0.63
10	0.62	0.55	499.9	295.8	5.1	3.7	2.3	2.3	1.8	5.3	12.1	2.8	2.8	2.5	0.68	0.66
20	0.61	0.51	500.9	293.4	4.4	3.1	2.5	2.4	1.7	4.9	12.1	2.7	2.4	2.1	0.72	0.69
30	0.61	0.45	420.2	286.7	4.0	2.9	2.6	2.6	1.7	4.3	10.1	2.6	2.2	1.9	0.75	0.74
40	0.59	0.26	373.1	252.1	3.8	2.8	2.7	2.8	1.7	2.5	9.0	2.4	2.1	1.9	0.79	0.80
50	0.57	0.18	353.8	244.4	3.7	2.6	2.9	3.0	1.6	1.8	8.5	2.3	2.1	1.7	0.85	0.86
80	0.45	0.14	302.5	167.3	3.5	2.0	3.3	3.5	1.3	1.3	7.3	1.6	1.9	1.3	0.97	1.00
100	0.38	0.12	268.0	143.6	2.5	2.0	3.4	3.5	1.1	1.1	6.5	1.3	1.4	1.3	1.00	1.00
150	0.35	0.11	171.1	129.1	2.1	1.5	3.4	3.5	1.0	1.0	4.1	1.1	1.2	1.0	1.00	1.00
200	0.35	0.10	41.4	104.7	1.8	1.5	3.4	3.5	1.0	1.0	1.0	1.0	1.0	1.0	1.00	1.00

ME: MARSE energy (s·V/GN); FL: flexibility (m/GN); PM: peak magnitude of wavelet transform; F: frequency of wavelet transform (kHz); NME: normalized MARSE energy; NFL: normalized flexibility; NPM: normalized peak magnitude; NF: normalized frequency.

**Table 2 sensors-21-02941-t002:** Maximum ratios between cavity and soil sections for each normalized value and estimated cavity size.

Normalized Value	Maximum Ratio		Estimated Cavity Size (mm)	
	T = 8 mm	T = 15 mm	T = 8 mm	T = 15 mm
MARSE energy	1.8	5.4	100	60
Flexibility	12.3	3.2	40	80
Peak magnitude	3.0	3.0	N/A	N/A
Frequency	0.63	0.63	80	80

## Data Availability

The data presented in this study are available on request from the corresponding author. The data are not publicly available due to ongoing research project.

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
