# Peer review of "Estimation of Cavities beneath Plate Structures Using a Microphone: Laboratory Model Tests"

_sensors, 2021, doi:10.3390/s21092941_

Round 1

Reviewer 1 Report

This paper proposed to detect the internal cavities by using a microphone. The authors conducted the laboratory test and analyzed the experimental data to demonstrate the feasibility of the detection using microphone. And the signal processing methods have been suggested in this paper. Though ultrasound could be a better option, the proposed study may provide a different application of low-frequency nondestructive testing. The paper itself is well-constructed. And the writing, figure and table in the manuscript are very clear and formal. Therefore, the paper is suggested to be published. But still, I have some questions as below:

  1. what about the range(depth) of detection by microphone?
  2. if the impact hammer is used to generate the sound, why aren’t other type of senor considered in the test? Such as PZT sensors, it has higher sensitivity than microphone I believe.
  3. Please provide specs of the microphone, e.g. frequency, sensitivity, SNR etc.
  4. some types of microphone can achieve very high resolution and sensitivity; however, the application frequency is very limited. Authors should discuss the merits of microphone over other sensors such as typical UT transducer.
  5. the waveguide may damp out the energy, which may introduce the SNR issue. Did author make the special design of the waveguide used in this paper?
  6. Did the authors consider the couplant between the tested object and microphone ? or just air coupled?

Reviewer 2 Report

The current study aims to determine the cavity location and size using sound waves in a chamber filled with sand. the authors measure the energy, flexibility, peak magnitude of wavelet transform, and frequency corresponding to the peak magnitude. The author scope is to highlight the accuracy of the cavity detection using sound waves. The authors reported that cavity detection is high but the cavity size estimation might be inaccurate.

Please consider reviewing the abstract and highlight the novelty, major findings and conclusions in more depth.

The introduction is very basic and simple, it does not reflect on past studies, what they have done and what were their major findings. Also, what does this study brings to the field in terms of novelty and knowledge.

What is the research gap did you find from the previous researchers in your field? Mention it properly. It will improve the strength of the article.

Line 229/230 please explain more in details the role of stiffness ratio and plate thickness, is this always the case in all past studies or does the medium have some effect on the relationship

Line 294 “due to the flexural vibration behavior of the upper plate” is this the only reason or there might be other reasons as well? Please mention them and support with references if there are any

Line 357 why do you add reference there? Are you referring to table 1 in both references? If not then this is misuse of citations, please remove and check elsewhere for this

Line 378 using maybe is not scientific either say is dependant or not? What did past studies say about this?

Line 405, again using maybe and over and under estimated, this means that the whole detection technique is not accurate at all. Mention the limitations of this method and how it can be improved

The results are merely described and is limited to comparing the experimental observation. The authors are encouraged to include more discussion and critically discuss the observations from this investigation with existing literature.

Reviewer 3 Report

The motivation of the manuscript is well described. Authors provided experimental study of cavity detection in laboratory experiment in order to simulate ground cavity under concrete pavements, retaining wall etc.  The approach is similar to search of cavity inside wall by knocking on the wall and listening, while here the human ear and finger are microphone and impact hammer, respectively. The paper is grammatically well-written and of minimum typos. The particular sections logically follow each other. Yet I have several remarks, which have to be addressed. I do not recommend the paper to be published in the recent form.   - The laboratory model is an ideal case. Beside the boundary conditions, the material and thickness of the plate drive the wave propagation and its attenuation in the plate. The authors have not addressed why they chose the material of the plate. Why the concrete or other material was not used ?   - The cavity in the experiment behaves as "speaker" with almost ideal interface between air and plastic cover, thus this speaker amplifies the sound of the impact. The different shape of the cavity will affect an ideal height of the microphone above the plate as well as other results of this paper. The authors have not addressed why they chose this shape of the cavity. Furthermore I would not agree with the first bullet paragraph in conclusions that "the impact energy is not transmitted to the soil but is trapped inside the acrylic plate.... the acrylic plate becomes more flexible at the cavity section due to the loss of support".   - Lamb waves are basically of two type considering the plates, which one was considered in your experiment ? - What was the precision of velocity measurement ? The measured velocity of Lamb waves is different from calculated ones. The wave speed is in the order of mm/us for free plate of 8 mm thickness and frequency in the order of kilohertz. - The transfer function of microphone,i.e. sensitivity dependence on frequency, should be given.   - Please specify the difference between particle velocity of the sound waves and phase and group velocities? - The equation (1) is not well-expressed formally since every mathematician/physics would say that speed [m/s] is not a time [s] - missing parentheses. - The conditions of experimental setups and evaluations are missing, sampling frequency, sampling points, mother wavelet type, etc.     - Please cite your own-work, which describes the problem. Even the paper is in your native language. Kang, S., Lee, J. S., Yu, J. D., & Kim, S. Y. (2020). Detection of Cavities Beneath Plate Structure using a Microphone. Journal of the Korean Society of Hazard Mitigation20(6), 229-237.

Round 2

Reviewer 2 Report

All questions were answered 

Reviewer 3 Report

Authors have addressed all my remarks.